# The MYC-YBX1 Circuit in Maintaining Stem-like Vincristine-Resistant Cells in Rhabdomyosarcoma

**DOI:** 10.3390/cancers15102788

**Published:** 2023-05-17

**Authors:** Madeline Fritzke, Kenian Chen, Weiliang Tang, Spencer Stinson, Thao Pham, Yadong Wang, Lin Xu, Eleanor Y. Chen

**Affiliations:** 1Department of Laboratory Pathology and Medicine, University of Washington, Seattle, WA 98195, USA; madi222@uw.edu (M.F.); tang98115@gmail.com (W.T.); spencer.stinson@oregonstate.edu (S.S.); phamqt11@gmail.com (T.P.); yadong@uw.edu (Y.W.); 2Quantitative Biomedical Research Center, Peter O’Donnell Jr. School of Public Health, University of Texas Southwestern Medical Center, Dallas, TX 75390, USA; kenian.chen@utsouthwestern.edu; 3Astellas US Technologies, Universal Cells, Inc., Seattle, WA 98121, USA

**Keywords:** rhabdomyosarcoma, stem cells, therapy resistance, MYC, YBX1, zebrafish

## Abstract

**Simple Summary:**

Rhabdomyosarcoma (RMS) is a devastating pediatric sarcoma. In particular, patients with relapsed disease have very poor survival outcomes. However, there has not been any significant change in therapy options for the last three or four decades, and the mechanisms underlying treatment failures remain poorly understood. In this study, we showed that two genes, *MYC* and *YBX1*, are essential for the viability of chemotherapy-tolerant cells that lead to resistance. We also demonstrated the mutual regulation of MYC and YBX1 function as a novel mechanism of therapy resistance in RMS. Targeting MYC, YBX1, and their interacting factors is a promising therapeutic approach to improve survival outcomes of RMS patients.

**Abstract:**

Rhabdomyosarcoma (RMS) is a pediatric soft tissue sarcoma that causes significant devastation, with no effective therapy for relapsed disease. The mechanisms behind treatment failures are poorly understood. Our study showed that treatment of RMS cells with vincristine led to an increase in CD133-positive stem-like resistant cells. Single cell RNAseq analysis revealed that MYC and YBX1 were among the top-scoring transcription factors in CD133-high expressing cells. Targeting MYC and YBX1 using CRISPR/Cas9 reduced stem-like characteristics and viability of the vincristine-resistant cells. MYC and YBX1 showed mutual regulation, with MYC binding to the YBX1 promoter and YBX1 binding to MYC mRNA. The MYC inhibitor MYC361i synergized with vincristine to reduce tumor growth and stem-like cells in a zebrafish model of RMS. MYC and YBX expression showed a positive correlation in RMS patients, and high MYC expression correlated with poor survival. Targeting the MYC-YBX1 axis holds promise for improving survival in RMS patients.

## 1. Introduction

Rhabdomyosarcoma (RMS) is a devastating pediatric soft tissue sarcoma (250–300 cases a year in the US) [1,2] and consists of two major subtypes based on genetic alterations: fusion-negative (FN) and fusion-positive (FP). FN RMS is characterized by mutations in the receptor tyrosine kinase/RAS/PIK3CA axis in greater than 90% of cases [3]. In contrast, FP RMS is characterized by a translocation event resulting in the fusion between *PAX3* or *PAX7* and *FOXO1* genes [4,5]. While localized disease of RMS can be controlled with surgery, chemotherapy, and radiation, there is no effective treatment against relapsed or metastatic disease regardless of subtype, with a less than 30% 3-year survival rate [6]. There is an urgent need to understand the biology underlying treatment failure to identify more effective therapy options for improving survival outcomes of treatment-refractory RMS.

To date, there are limited data on the molecular basis of chemotherapy resistance in RMS. A previous study demonstrated the survival of a cell subpopulation expressing *MYOD1* in an RMS cell line following treatment with the standard-of-care chemotherapeutic agents vincristine and doxorubicin [7]. Other studies have shown high levels of antiapoptotic proteins are present in treatment-refractory RMS tumors [8,9] and upregulation of selective signaling pathways (e.g., GP130/STAT and Hedgehog) in chemotherapy-resistant RMS cells [10,11]. Recently, a study by Patel et al. demonstrated that chemotherapy eliminates proliferative myoblast-like cells in the FN embryonal subtype of RMS (ERMS), leaving behind an immature cell population that recapitulates paraxial mesodermal cells in development [12]. A study by Danielli et al. showed the chemotherapy eliminated cycling myogenic progenitor cells and enriches for muscle stem-like cells in the FP alveolar RMS (ARMS) [13]. Overall, previous findings suggest that there are transcriptional changes that alter cell state in proliferation and differentiation as RMS cells develop therapy resistance. However, the cellular and molecular mechanisms underlying early cell state adaptation leading to the emergence of chemotherapy-resistant RMS cells remain unknown.

In this study, we aimed to characterize the cell state and molecular mechanisms underlying the biology of therapy-resistant RMS cells in response to treatment with the standard-of-care chemotherapeutic agent vincristine.

## 2. Results

### 2.1. Vincristine Treatment Generated a Population of RMS Cells with Stem-like Characteristics

To characterize the cellular characteristics and state of vincristine-resistant RMS cells, we treated a panel of FN and FP RMS cell lines (FN: RD and SMS-CTR; FP: Rh5 and Rh30) with a concentration of vincristine that killed 80–90% of cells (IC 80–90) over 7 days. This treatment resulted in a population of slow-growing cells in comparison to the parental lines (Figure 1A–E; Appendix A). These vincristine-resistant cells showed enlarged and irregular cell sizes with bizarre-shaped nuclei (Figure 1B,D). By cell cycle analysis, vincristine-resistant cells showed a delay in the cell cycle, with an increased number of cells arrested in the G2/M phase compared to the parental cells (Figure 1F).

We then assessed whether vincristine treatment altered the cell state of the RMS cells, and we showed that in comparison with the parental cells, FN RD and SMS-CTR cells following 7-day treatment with vincristine at IC 80–90 were enriched with CD133-positive cells by immunofluorescence (IF) (Figure 1G–J) and flow cytometry (Figure 1K; Appendix A). These cells also showed statistically significant upregulation of the stem cell markers *CD133, PAX7*, *OCT4*, *NANOG* and *SOX2* by quantitative RT-PCR (unpaired *t*-test, *p* < 0.01, Figure 1L).

### 2.2. FN RMS Spheres Enrich for CD133-Expressing Stem-like Cells and Show Increased Resistance to Standard-of-Care Therapeutics

RMS spheres cultured in stem-cell media have previously been shown to be enriched for CD133-positive stem-like cells and are more resistant to the chemotherapeutics cisplatin and chlorambucil [14]. Similarly, we showed that serially passaged spheres derived from RD showed an increase in CD133 expression as well as CD133-expressing cells compared to the bulk xenograft tumor derived from the RD line (ANOVA, *p* < 0.0001, Appendix A). The spheres generated from the SMS-CTR cells also showed a similar increase in the CD133-expressing cells (Appendix A). By contrast, the spheres generated from the two FP RMS lines (Rh5 and Rh30) lacked CD133-positive expressing cells (Appendix A), suggesting that a different marker may better highlight the stem cell-like population in FP RMS cells. However, the spheres generated from both FN RMS cell lines (RD and SMS-CTR) and FP RMS cell lines (Rh5 and Rh30) treated with varying doses of the standard-of-care chemotherapeutic agent, vincristine, all exhibited decreased sensitivity compared to the parental adherent cells, as evidenced by an ~3- to 9-fold increase in IC50 (Appendix A).

### 2.3. MYC- and YBX1-Driven Transcriptional Regulatory Networks Are Upregulated in CD133-Positive Stem-like RMS Cells

Given the technical challenge of getting a sufficient number of intact vincristine-treated RMS cells for single-cell RNA sequencing analysis, we used serial passaging of RD sphere cells cultured in stem cell media to enrich for CD133-positive cells in order to perform gene expression profiling by single-cell RNA sequencing. Sphere cells harvested at passages 1 (P1) and 5 (P5) were subjected to processing and analysis by single-cell RNA sequencing. As displayed by Uniform Manifold Approximation and Projection (UMAP) (Figure 2A), both P1 and P5 RD sphere cells showed similar clustering patterns (clustering for P1 and P5 shown in Figure 2A), with approximately 19 unique clusters identified. By gene annotation analysis, cluster 3 is noted for genes involved in skeletal muscle development and structure (e.g., *ACTC1*, *TTN*, *MYLPF,* and *MYL1*); cluster 12 is noted for genes involved in extracellular matrix and cell migration (e.g., *COL1A1*, *COL1A2*, and *FN1*); and cluster 14 is noted for genes involved in cell adhesion and cytoskeletal interactions (e.g., *RND3* and *VIM*) (see heatmap of top differentially expressed genes in Appendix A). *CD133* (*PROM1*)-expressing cells showed a relatively diffuse distribution throughout all clusters (Figure 2B), indicating dynamic transcriptional plasticity of CD133-expressing cells. Through enrichment analysis against the TRRUST (Transcriptional Regulatory Relationships Unraveled by Sentence-based Text mining) database using Metascape [15], we observed that MYC and YBX1 are among the top-scored master transcription factors in *CD133*-high expressing RD cells compared to *CD133*-low expressing cells (Figure 2C). Protein expression of MYC and YBX1 was also detected in serially passaged spheres by immunohistochemistry (RD P5 spheres shown in Figure 2D), with MYC showing nuclear localization and YBX1 showing cytoplasmic localization. In addition, we found stronger correlations between MYC and YBX1, MYC and MYC target genes and between YBX and YBX1 target genes in CD133-high-expressing sphere cells than in CD133-low-expressing cells (Figure 2E; Appendix A). Upregulation of well-known downstream target genes of MYC and YBX1 was noted in RD cells with high CD133 expression (Figure 2F), which was independently validated in RD sphere cells using quantitative RT-PCR (Appendix A).

### 2.4. Targeted Disruption of MYC and YBX1 Inhibits Tumor RMS Tumor Cell Growth and Sphere Formation and Induces Cell Death in CD133-Positive Stem-like Cells

Given the strong positive correlation between MYC and YBX1 as well as their corresponding target genes in CD133-high FN RD cells, we decided to focus on characterizing the loss-of-function effects of MYC and YBX1 on RMS tumor growth and stem cell function as well as their potential interactions. To assess the loss-of-function effects of MYC and YBX1 on RMS tumor cell growth and sphere formation, FN RMS (RD, SMS-CTR) and FP RMS (Rh5 and Rh30) Cas9-expressing stable lines were transduced with lentiviral constructs expressing double gRNAs (dgRNA) against *MYC* and *YBX1*. CRISPR/Cas9-mediated disruption of *MYC* and *YBX1* (Figure 3A; Appendix A) inhibited cell growth according to cell counts and CellTiter-Glo viability assays (Figure 3B; Appendix A) and reduced sphere formation frequency and size at two cell dilutions (Figure 3C–G; Appendix A). To demonstrate gene targeting specificity, we showed that overexpression of Cas9-resistant MYC and YBX1 alleviated the cell growth defect resulting from CRISPR/Cas9-mediated gene targeting of *MYC* and *YBX1*, respectively (Appendix A). By PCR using the primers flanking the double gRNA sites, we also showed gene-specific DNA deletion events for *MYC* and *YBX1* (Appendix A).

We generated a CD133:GFP SMS-CTR reporter line by knocking in the GFP cassette in frame with the CD133 locus via CRISPR/Cas9 in order to visualize CD133-positive cells in culture. To assess the effect of *MYC* and *YBX1* gene disruption on the viability of CD133-positive sphere cells in real time, we performed live imaging of dissociated sphere cells derived from the CD133:GFP SMS-CTR stained with the fluorogenic NucView^®^ 530 Caspase-3 substrate 5 days following lentiviral transduction of the CRISPR dgRNA constructs and antibiotic selection. We observed an increased number of cells undergoing apoptosis, including CD133-positive cells (indicated by arrowheads in Figure 3H; quantitation in Appendix A, ANOVA, *p* < 0.001). Together, *MYC*- and *YBX1*-targeted disruption decreased the stem-like characteristics of RMS cells and contributed to increased apoptosis in the CD133-positive cell population, at least in FN RMS.

### 2.5. MYC and YBX1 Play an Important Role in Modulating the Chemosensitivity of RMS Cells

Given that MYC and YBX1 are essential for the viability of CD133-positive stem-like FN RMS cells and that short-term vincristine-treated FN RMS-resistant cells showed an increase in CD133-positive stem-like cells, we next asked whether MYC and YBX1 play a role in modulating the chemosensitivity of FN RMS cells to vincristine. We first showed, by quantitative RT-PCR, that there was a relative increase in the expression of *MYC* and *YBX1* in the resistant RD and SMS-CTR cells following 10-day treatment with vincristine at IC 80–90 compared to vehicle (DMSO)-treated parental cells (unpaired *t*-test, *p* < 0.01, Figure 4A). To assess whether YBX1 and MYC are essential for the viability of resistant FN RMS cells following short-term treatment with vincristine at IC 80–90, we transduced Cas9-expressing RD cells that survived 10 days of treatment with vincristine at IC 80–90 with gene-specific double gRNAs (dgRNAs) and restarted vincristine treatment following transduction and antibiotic selection. While the vincristine-treated cells with targeted disruption of a safe-harbor genomic region targeting as a control remained relatively tolerant to vincristine compared to the parental cells, targeted disruption of *MYC* and *YBX1* reduced cell growth and increased cell death in the vincristine-treated cells (ANOVA, *p* < 0.05, Figure 4B–D and Appendix A). Overall, the findings indicate that MYC and YBX1 are essential for maintaining the viability of drug-tolerant FN RMS cells in response to short-term vincristine treatment.

### 2.6. Reciprocal Regulation of MYC and YBX1 in FN RMS Cells

Given the role of MYC in gene transcription and the DNA and RNA-binding capacity of YBX1, we next asked whether MYC and YBX1 showed mutual regulation at the transcriptional or posttranscriptional level. We first observed that upon targeted disruption of *MYC* in two FN RMS lines, RD and SMS-CTR, there was reduced mRNA expression of *YBX1* by quantitative RT-PCR and protein expression by Western blot analysis (Figure 3A, Appendix A and Figure 5A,B). Similarly, targeted disruption of *YBX1* in RD and SMS-CTR cells also reduced *MYC* mRNA and protein expression (Figure 3A and Figure 5A,B). The findings suggest that there might be mutual regulation of MYC and YBX1 at the transcriptional and/or post-transcriptional level. We first tested whether MYC regulated the expression of *YBX1* at the level of gene transcription by performing the CUT and RUN assays on RD and SMS-CTR cells treated with vincristine. MYC bound to the promoter containing the consensus MYC binding motif (CACGTG) of YBX1, CD133 and muscle satellite/RMS tumor propagating cell markers MYF5 and PAX7 (Figure 5C,D). YBX1 was expressed in the cytoplasm in RMS cells (Figure 2D), suggesting its functional role at the posttranscriptional level. To determine whether YBX1 binds to *MYC* mRNA, we demonstrated the YBX1 protein-*MYC* mRNA interaction by an RNA immunoprecipitation assay (RIP) performed on RD cells using an antibody against YBX1 in the RNA pull down and primers against the 3′ UTR region of *MYC* mRNA (Figure 5E). Overall, the findings indicate that MYC and YBX show reciprocal regulation, with MYC regulating the gene expression of *YBX1* and YBX1 regulating the mRNA stability of *MYC*.

### 2.7. Combination Treatment with the MYC Inhibitor MYCi361 and Vincristine Significantly Inhibited RMS Tumor Cell Growth and Depleted the Stem-like Cell Subpopulation In Vivo

MYCi361 is a MYC inhibitor previously described by Han et al. that decreases MYC protein activity by disrupting MYC/MAX dimer formation and impairing MYC-driven gene expression [16]. We have demonstrated so far that MYC plays an essential role in maintaining the stem cell-like vincristine tolerant cells, and we next asked whether the combination treatment of MYCi361 and vincristine could exert a more significant effect on inhibiting RMS growth in comparison to the single-agent treatment alone. We showed in cultured RD and SMS-CTR cells that treatment with the combination of MYCi361 and vincristine significantly reduced cell growth compared to treatment with each agent alone (Appendix A). By quantitative RT-PCR, MYCi361-treated RD and SMS-CTR cells showed decreased expression levels of known MYC target genes (Appendix A). To determine whether MYCi361 could be a potential therapeutic agent in combination with vincristine to reduce RMS tumor cell growth in vivo, we tested the two-agent combination in the KRAS(G12D)-induced zebrafish FN RMS model. Primary zebrafish RMS tumor cells expressing a transgenic (*rag2*:dsRed) fluorescent reporter derived from three tumors were transplanted into a pool of syngeneic CG1-strain host zebrafish, and engrafted tumor fish were treated with DMSO (vehicle control), MYCi361 (100 mg/kg), vincristine (0.4 mg/kg) or MYCi361 and vincristine in combination with half the dosage for each drug for 7 days. The dosage of MYCi361 and vincristine was titrated to a level that only slightly affected tumor growth as a single agent in vivo. We observed that treatment of zebrafish RMS tumors with the combination of vincristine and MYCi361 significantly reduced tumor growth compared to the tumors treated with vincristine or MYCi361 alone (Figure 6A,B). This was supported by a significantly greater decrease in the proliferative index and increase in the number of apoptotic cells in tumors treated with the two-drug combination, as highlighted by immunohistochemistry for Ki67 and Caspase-3, respectively (see representative tumor sections from each treatment cohort and quantitative results in Figure 6C,D).

To determine whether MYCi361 modulates the relative proportions of tumor cell subpopulations based on their differentiation status, we took advantage of the *myf5:GFP/mylz2:mCherry* transgenic zebrafish line to label distinct cell subpopulations of FN RMS based on their differentiation status in vivo [17]. The *myf5*:GFP+/*mylz2*:mCherry-negative (G+R-) stem cell-like tumor-propagating-cell (TPC) population has been shown to have self-renewal capacity, while late-differentiating *myf5*:GFP-negative/*mylz2*:mCherry+ tumor cells (G-R+) lack the capacity to self-renew [17,18]. Following 7-day drug treatment of fish bearing engrafted *myf5*:GFP/*mylz2*:mCherry-expressing RMS tumors, the relative proportions of cell subpopulations were assessed by fluorescence-activated cell sorting (FACS) (n = 3 independent tumors). Treatment with MYCi361 resulted in significant depletion of the proportion of the *myf5*:GFP+/*mylz2*:mCherry-negative TPC population and a concomitant increase in the proportion of *myf5*:GFP-negative/*mylz2*:mCherry+ late-differentiating cells (Representative flow cytometric analysis and quantitation of cell subpopulations in Figure 6E,F; *p* < 0.05). MYCi361-treated tumor cells also showed decreased expression of the stem genes *pax7* and *myf5* (Appendix A).

### 2.8. High MYC Expression Correlates with Poor Survival Outcomes of FN RMS

To translate our findings into clinical relevance to RMS patients, we first determined whether MYC and YBX1 expression could be detected in recurrent or metastatic RMS samples. We analyzed the protein expression of MYC and YBX1 by immunohistochemistry in a tissue microarray of tissue cores from 32 RMS patients, including 17 primary tumors (12 FN and 5 FP) and 15 recurrent or metastatic tumors (8 FN and 7 FP) along with 5 muscle controls. FN samples included ERMS (*n* = 9), FN ARMS (*n* = 2), and spindle cell/sclerosing variants (*n* = 1). MYC and YBX1 were expressed in at least a subset of primary RMS samples (MYC: 3 of 9 (33%) in FN RMS, 1 of 5 (20%) in FP RMS; YBX1: 12 of 12 (100%) in FN RMS, 5 of 5 (100%) in FP RMS) as well as recurrent/metastatic RMS samples (MYC: 4 of 8 (50%) in FN RMS, 1 of 7 (14.3%) in FP RMS; YBX1: 8 of 8 (100%) in FN RMS, 7 of 7 (100%) in FP RMS). MYC staining showed a patchy distribution, while YBX1 staining showed a diffuse distribution (Figure 7A). All normal muscle tissue samples were negative for both MYC and YBX1 (Figure 7A). Overall, MYC showed a trend of increased frequency of positive expression in recurrent/metastatic FN RMS, and YBX1 was expressed in all primary and metastatic FN and FP RMS tumor tissue samples examined (Figure 7B), suggesting that MYC and YBX1 play a role in both primary and recurrent/metastatic RMS tumors.

To examine the potential roles of *MYC* and *YBX* as biomarkers for RMS patients, we studied *MYC* or *YBX* mRNA expression in 81 RMS cases with survival data (63 fusion-negative and 18 fusion-positive). We first showed that high expression of *MYC* correlated positively with high expression of *YBX1* in both FN and FP patients (Figure 7C,D). We then asked whether *MYC* or *YBX1* expression in RMS tumors was associated with patient survival. Kaplan–Meier curves were generated based on gene expression values dichotomized into over- and under-expressed groups using the median expression value within each cohort as a cutoff. However, while YBX expression did not correlate with overall survival, high MYC expression correlated with decreased overall survival in FN RMS patients (Figure 7E, log-rank test, *p*  =  0.043, HR  =  2.08, 95% CI  =  1.04–4.51) but not FP RMS patients (Figure 7F, *p*  =  0.673, HR  =  1.46, 95% CI  =  0.25–8.48). Although the small sample size in both subsets compromises the ability to draw firm conclusions, the findings suggest that the expression of *MYC* might be useful as a clinical prognostic biomarker if they are confirmed in a prospective analysis.

## 3. Discussion

MYC is a regulatory transcription factor with essential roles in regulating embryonic stem cell identity and various aspects of cancer biology, including proliferation, growth, metabolism and differentiation [19,20,21,22]. MYC has also been shown in a few cancer types as one of master transcription factors to control cell phenotypes or identities. For example, in breast cancer, MYC-driven epigenetic reprogramming causes activation of de novo enhancers to support dedifferentiation and onset of a stem-like state while repressing transcriptional activity of lineage-specifying transcription factors [19]. In pancreatic ductal adenocarcinoma, MYC binds to neuroendocrine genes to facilitate ductal-neuroendocrine plasticity and thereby contributes to chemotherapy resistance [23]. Our study demonstrated that MYC binding to the E-box DNA response element within the promoters/enhancers of stem genes in FN RMS and disruption of MYC activity by either CRISPR/Cas9 gene targeting or treatment with the MYC inhibitor MYCi361 reduced stem-like characteristics and increased apoptosis of vincristine-resistant FN RMS cells. These findings suggest that MYC-driven transcriptional regulation plays an important role in maintaining stem cell state and modulating the state plasticity of resistant cells in response to treatment with vincristine.

The study by Patel et al. demonstrated that RMS tumors mimic the spectrum of embryonic myogenesis. After chemotherapy treatment, the immature population that resembles embryonic paraxial mesoderm expands to replace proliferative myoblast-like cells. Together, chemotherapy resistance in RMS likely arises from cell state plasticity in treated tumor cells with a shift towards a more stem-like or immature embryonic phenotype. However, the molecular mechanisms driving the dynamics of cell state plasticity over time in response to vincristine treatment and how MYC and other transcription factors contribute to this process still require further investigation.

MYC activity has been shown to either promote apoptosis or survival of cancer cells in response to an anti-mitotic agent. In response to taxol, MYC up-regulates expression of pro-apoptotic genes to induce cell death of breast, colon, ovarian, and lung cancer cells [24]. By contrast, unphosphorylated MYC binds to the microtubules of the mitotic spindle following vincristine treatment of diffuse large B cell lymphoma cells, allowing for transfer of functional MYC to dividing daughter cells, contributing to vincristine resistance and induction of polypoidy [25]. In our study, RMS cells showed increased expression of MYC following a short-term treatment with vincristine treatment, and targeted disruption of *MYC* induced apoptosis in vincristine-tolerant RMS cells. How MYC regulates the balance of pro- and anti-apoptotic gene activity in the stem cell-enriched vincristine tolerant RMS and whether MYC-microtubule interaction promotes tolerance or resistance to vincristine remain to be further investigated.

Our study demonstrated that MYC and YBX1-driven transcriptional regulatory networks were enriched in the CD133-high stem cell population in FN RMS. YBX1 is a DNA- and RNA-binding protein with diverse functions in transcriptional and posttranscriptional regulation [26,27]. In our study, targeted disruption of *YBX1* also phenocopied the loss-of-function effects of MYC on the viability of the CD133-positive RMS stem cell population and the stem cell-enriched vincristine-resistant cells. Mechanistically, MYC promoted the transcriptional activity of *YBX1* by binding to the E-box motif in the *YBX1* promoter. YBX1 in turn binds to *MYC* mRNA to regulate its stability. To date, only a few studies have characterized the MYC-YBX1 interaction in cancer. The mutual regulation of the MYC-YBX1 circuit has only been described so far in multiple myeloma, where the perturbation of the MYC-YBX1 circuit induces apoptosis in multiple myeloma cells [28]. YBX1 has been shown to either promote or decrease *MYC* mRNA stability. Specifically, YBX1 binds to a long noncoding RNA (Linc2042) in squamous cell carcinoma to facilitate tumor cell growth and metastasis [29]. In contrast, YBX1 promotes mRNA decay of *MYC* in an m6A-dependent manner to maintain the viability of acute myeloid leukemia cells [30]. In our study, the regulatory loop of MYC-YBX1 resulted in increased *MYC* mRNA stability and *YBX1* gene expression, which is a novel mechanism for maintaining the viability of stem-like vincristine-resistant cells. Whether the MYC-YBX1 circuit plays a role in the progression to therapy resistance in other cancer types remains to be further investigated.

In our study, targeted disruption of *MYC* and *YBX1* reduced the growth and self-renewal frequency of stem-like sphere cells in both FN and FP RMS cells. However, as our study identified the MYC and YBX gene modules initially from the CD133-high stem cell-like population in FN RMS, and subsequent regulatory interactions between MYC and YBX were also demonstrated only in FN RMS, it is likely that some other pathways in addition to or instead of the MYC-YBX1 axis contribute to the regulation of stem cell identity and therapy resistance in FP RMS. YBX1 has previously been shown to promote the invasion and metastasis of several sarcoma types, including FN and FP RMS, by binding to the 5′ untranslated region of HIFα mRNA and activating its translation [31]. The post-transcriptional activity of YBX1 plays a role in various aspects of tumor growth and progression in both FN and FP RMS. Further investigations will be necessary to determine how YBX1 functions differently between FN and FP RMS.

MYC, similar to many other transcription factors, lacks a deep surface binding pocket, thereby making drug design challenging. To date, the MYC-derived bHLH-zipper domain peptide Omomyc and JQ1, a small molecular inhibitor of the bromodomain and extraterminal (BET) protein BRD4, have been shown to have a robust capacity to suppress the transcriptional activity of MYC [32,33]. In this study, we showed that the small molecule MYC inhibitor MYCi361, which was recently shown by Han et al. to directly inhibit MYC activity [16], sensitized FN RMS cells to vincristine treatment in vitro and in a zebrafish model in vivo. The combination of MYCi361 and vincristine significantly reduced RMS tumor growth and the stem cell-like tumor propagating cell population. In addition, there was a positive correlation between *MYC* and *YBX1* expression levels in RMS patients, and high *MYC* expression levels correlated with poorer survival outcomes in our analysis of 81 RMS patients. Our findings indicate that targeting MYC or MYC-driven transcriptional networks is a promising therapeutic option for reducing the likelihood of chemotherapy resistance in RMS and may be a useful clinical prognostic biomarker if our findings are confirmed in a larger-scale prospective analysis.

A direct YBX1 inhibitor, SU056, has recently been shown to inhibit the cell proliferation of ovarian cancer cells by modulating YBX1-associated proteins and pathways, and the inhibition of YBX1 activity by SU056 treatment sensitizes ovarian cancer cells to palictaxel [34]. In depth characterization on the effects of SU056 on RMS growth and disease progression, as well as an investigation of whether SU056 treatment mimics the loss-of-function effects of YBX1 on stem cell characteristics and therapy resistance in RMS, will be part of our future investigations.

We should note some limitations of the study. Due to the scarcity of patient tumors that have undergone recent treatment with the vincristine and the challenges of obtaining intact cells from RMS cell lines post-vincristine treatment, we employed stem cell-rich RMS spheres as a substitute for single cell RNA sequencing analysis. This surrogate method offers some understanding of the expression patterns of CD133-high cells in the spheres, but it may not completely reflect the expression patterns of the stem cell-like CD133-high cell population after vincristine treatment. In addition, our survival analysis was based on a limited sample of RMS patients with available expression data. Further studies with a larger and more diverse patient population are needed to improve the robustness of the results. It is also important to consider potential confounding factors, such as treatments and co-morbidities. In addition, in the rescue experiment to assess CRISPR/Cas9-mediated gene targeting specificity, we demonstrated that overexpression of Cas9-resistant MYC and YBX1 alleviated the cell growth defect resulting from CRISPR/Cas9-mediated gene targeting of *MYC* and *YBX1*, respectively (Appendix A), but there was no complete rescue. This could be due to insufficient overexpression of the Cas9-resistant MYC and YBX1 constructs or background CRISPR/Cas9-mediated off-target events [35].

## 4. Conclusions

In summary, our study has shown that vincristine treatment of FN RMS cells resulted in a stem-like resistant population that showed upregulation of genes regulated by MYC and YBX1. The MYC-YBX1 mutual regulatory circuit contributed to the viability and maintenance of stem-like vincristine-resistant cells. Targeting the MYC-YBX1 axis is a promising therapeutic option for reducing therapy resistance, at least in FN RMS.

## 5. Methods

### 5.1. CRISPR/Cas9 Gene Targeting in Human RMS Cell Lines

Single knockout was accomplished by transducing RMS cells with lentiviral virus expressing safe-harbor control [36] or gene-specific double gRNAs (Appendix A) and Cas9. Cells transduced with lentivirus were plated for cell-based assays following antibiotic selection 5 days post-transduction. Cloning of Cas9 and gRNA expression vectors was performed as described previously [36].

The coding portions of *MYC* and *YBX1* were amplified from the plasmids [37] obtained from Addgene (Watertown, MA, USA) for cloning purposes. Silent mutations to alter PAM sites to create a Cas9-resistant MYC lentiviral overexpression construct used in rescue and overexpression experiments were introduced using a 4-piece Gibson cloning strategy. YBX1 gRNAs were designed against noncoding regions of *YBX1*, so the coding portion was not altered in the lentiviral *YBX1* overexpression construct. All cell lines were previously authenticated by STR profiling and tested for mycoplasma contamination.

### 5.2. Single Cell RNA Sequencing (sci-RNA-seq3 Protocol)

RMS spheres were fixed in 4% paraformaldehyde and frozen in liquid nitrogen prior to nuclear extraction. The samples were subsequently submitted to the single-cell services at the Brotman Baty Institute for Precision Medicine (University of Washington) for single-cell RNA library preparation and sequencing.

### 5.3. Preprocessing of Single-Cell RNA Sequencing Data

Base calls from sequencing were converted into fastq format using Illumina’s bcl2fastq program and demultiplexed by the University of Washington sequencing core. Reads were adaptor-clipped using trim_galore with default settings. Trimmed reads were mapped to the human reference genome (GRch38) using the STAR program (version 2.5.2b). Uniquely mapping reads were extracted, and duplicates were removed based on unique molecular identifier (UMI) sequences. To generate expression matrices, the number of UMIs for each cell mapping to the exonic and intronic regions of each gene was calculated. Potential ambient RNA reads were estimated and removed using the R package SoupX [38]. Doublets were then identified using the Python package Scrublet [39]. Further analysis for quality filtering was performed using the Seurat R package (version 3.2.2). Only cells with a total read count <10,000 and number of genes detected >100 were kept. To remove potential dead cells from the analysis, cells with >15% mitochondrial reads were filtered out. In total, we obtained 82,133 cells from 4 conditions.

### 5.4. Integrated Analysis of Single Cell RNA Datasets

To account for potential batch effects, we performed an integrated analysis using the IntegrateData function implemented in the Seurat package (version 3.2.2). Briefly, we performed data normalization using the NormalizeData function and identified the top variable genes using FindVariableFeatures. We then selected integration features and anchors using the SelectIntegrationFeatures and FindIntegrationAnchors functions, respectively. Finally, we passed the anchors to the IntegrateData function and used dimensions of 50 to generate integrated Seurat objects.

### 5.5. Visualization and Clustering

For data visualization, we used Uniform Manifold Approximation and Projection (UMAP) to project cells onto 2D space. Using graph-based clustering, we clustered the cells into 20 clusters using the FindCluster function in Seurat with a resolution of 0.3.

### 5.6. Imputation of Single Cell RNAseq Data and Enrichment Analysis

To categorize cells into CD133 high and CD133 low cells, we used Markov affinity-based graph imputation of cells (MAGIC) to denoise the cell count matrix and fill in missing values to overcome the dropout problem that is faced by current single-cell technology. According to the imputed CD133 expression levels, cells were designated as CD133 high/low cells if their expression level was in the top 1/3 or bottom 1/3 of the condition (P1/P5), respectively. Differentially expressed genes comparing CD133-high and CD133-low cell populations were identified using the Wilcoxon rank sum test implemented in the Seurat FindMarkers function.

### 5.7. Drug Treatment Studies and Assessment of Tumor Growth Using the Zebrafish RMS Model

Zebrafish were maintained in a shared facility at the University of Washington under protocol #4330-01 approved by the University of Washington Subcommittee on Research Animal Care. Primary tumors *rag2*:*KRASG12D-T2A-GFP* were expanded by transplantation into 3–4 syngeneic adult CG1 fish. Tumor cells were then harvested by dissection and dissociation. For drug treatment studies, tumor cells were first transplanted subcutaneously into the peritoneal cavity at 20,000 cells per fish. A drug treatment study was performed on engrafted tumor-bearing tumor fish ~10 days post-transplantation for 7 days. Drugs reconstituted in DMSO, vincristine (Sigma Aldrich) at 8 mg/kg and MYCi361 (100 mg/kg), and vehicle (DMSO) were delivered on day 0 and day 4 in a 5-microliter volume using a Hamilton syringe needle via the intraperitoneal route. Imaging of tumor-bearing fish at day 0 and day 7 was performed using a Nikon fluorescent dissecting scope. The tumor volume change was quantified using ImageJ software as previously described [40].

### 5.8. Cell-Based Assays

Cell counts, CellTiter-Glo assays, sphere assays and flow cytometry assays were performed based on previously described protocols (18, 39). (See additional details in the Appendix A).

### 5.9. Flow Cytometry

To prepare live CD133-stained cells for flow cytometry, cultured RD and SMS-CTR cells were trypsinized, collected into conical tubes, and spun down in the centrifuge at 2000× *g*. The cell pellets were washed with Phosphate Buffered Saline (PBS) and resuspended in 0.5% Bovine Serum Albumin (BSA)/PBS. The cells were then stained with the PE-conjugated CD133 antibody (1:50; Clone TMP4; Thermo Fisher Scientific, Waltham, MA, USA) in the dark for 15 min at room temperature. Following the incubation, the cells were spun down at 2000× *g* and resuspended in cold PBS for subsequent flow cytometry analysis. Parental cells with no antibody treatment and serially passaged sphere cells stained with PE-conjugated CD133 were used as negative and positive gating controls, respectively. To determine the fold change in the enrichment for CD133-positive cells, the ratio of the percentage of CD133-positive cells in the vincristine-treated cells over the percentage in the DMSO (vehicle-treated) parental cells was calculated.

### 5.10. CUT&RUN Assay

The CUT&RUN assay was performed using the EpiNext^TM^ CUT&RUN Fast Kit per the manufacturer’s instructions. Approximately 10,000 cells were used for each reaction. The rabbit monoclonal MYC antibody (clone E5Q6W; Cell Signaling, Danvers, MA, USA) and nonimmune IgG as supplied in the kit were used in the chromatin capture reaction. The primers used for PCR verification are listed in Appendix A.

### 5.11. RNA-Binding Protein Immunoprecipitation Assay

The RNA-binding protein immunoprecipitation assay was performed using the Magna RIP kit (Millipore) per the manufacturer’s instructions. Rabbit polyclonal anti-YBX1 (1:100; clone D299; Cell Signaling, Danvers, MA, USA) and negative control rabbit IgG were used in the immunoprecipitation step. The primers used to detect MYC mRNA transcripts are listed in Appendix A.

### 5.12. Survival Association Analysis in the RMS Patient Cohort

The 81 RMS cases with survival and gene expression data were published previously [41]. A Cox proportional hazards model was used to calculate the statistical significance, hazard ratios and 95% confidence intervals of the associations between gene expression and survival. Kaplan–Meier curves were generated based on gene expression values dichotomized into over- and under-expressed groups using the within cohort median expression value as a cutoff.

### 5.13. Statistics

Two-tailed *t*-test and one-way/two-way ANOVA tests were used to assess statistical significance in differences between experimental and control samples when appropriate. A *p* value < 0.05 was considered statistically significant.

Additional details for immunohistochemistry, immunofluorescence, Western blots, and quantitative RT-PCR are provided in the Appendix A. All uncropped blots are inserted into Appendix A and quantitation of raw western blots densitometry is in Appendix A.

## Figures and Tables

**Figure 1 cancers-15-02788-f001:**
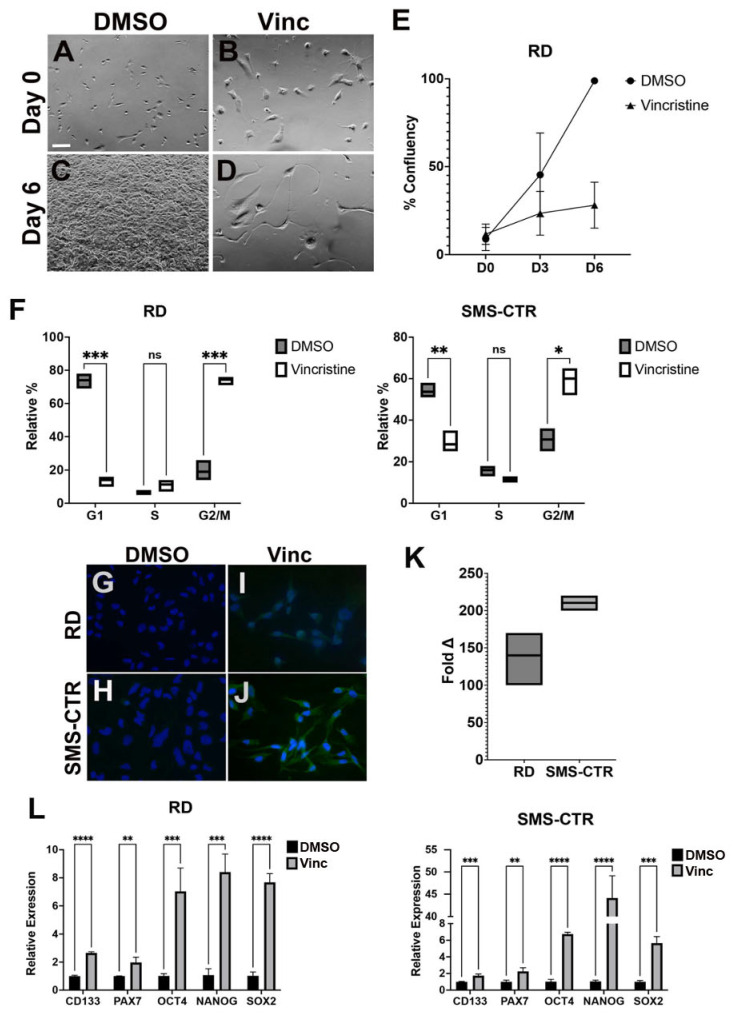
RMS-resistant cells in response to vincristine treatment show stem-like characteristics. (**A**–**D**): Representative bright-field images of RD cells treated with DMSO (vehicle) and continuous exposure of vincristine at 1 nM on day 0 (**A**,**B**) and day 6 (**C**,**D**). Magnification: 100×. Scale bar: 100 microns. (**E**) The changes in the percentage of confluence for the RD cells treated over 6 days as quantified using the ImageJ software. Shown here are the results from three independent experiments. (**F**) RD and SMS-CTR cells treated with vincristine at IC 80–90 (1 nM) were fixed and stained with DAPI for cell cycle analysis by flow cytometry. Quantitative summary from three independent experiments is shown. The line indicates the mean. (**G**–**J**) Representative images of immunofluorescence (IF) against CD133 on RD and SMS-CTR DTPs following 10 days of vincristine treatment. DMSO vehicle-treated cells in (**G**,**H**), and vincristine-treated ERMS cells (2 nM for RD and 1.5 nM for SMS-CTR) in (**I**,**J**). FITC: CD133 staining; Blue: DAPI nuclear staining. (**K**) Fold enrichment compared to the parental cells as determined by flow cytometry analysis of CD133 antibody-labeled RD and SMS-CTR cells. The line is the mean from three independent analyses. (**L**) Quantitative RT-PCR on RD and SMS-CTR cells treated with either DMSO or vincristine (1 nM) for 7 days. Error bar: standard deviation (STD) of 3 replicate samples. * = *p* < 0.05; ** = *p* < 0.01; *** = *p* < 0.005; **** = *p* < 0.0005 and ns = no significance by unpaired *t*-test (**F**,**L**).

**Figure 2 cancers-15-02788-f002:**
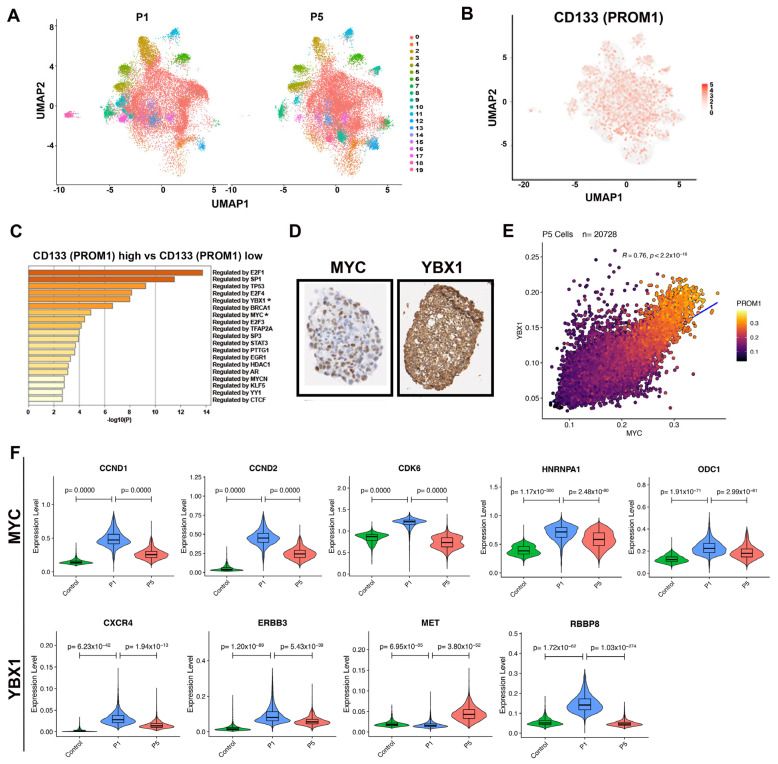
Single cell RNA sequencing expression analysis reveals MYC and YBX1 top regulators in CD133-high expressing ERMS cells. (**A**) UMAP demonstrates clustering of RD spheres from passage 1 (P1) and passage 5 (P5). (**B**) UMAP analysis for *CD133* (*PROM1*)-expressing cells in the RD sphere cells. (**C**) TRUUST analysis for top transcription factor-interacting regulators in *CD133* (*PROM1*) high-expressing sphere cells. * indicates *MYC* and *YBX1*. (**D**) Immunohistochemistry assessing MYC and YBX1 protein expression in RD spheres from P5. (**E**) Pearson correlation analysis on the relationship between MYC and YBX1 in RD sphere cells from P5. Heatmap indicates expression levels of CD133 in RD cells analyzed. (**F**) Violin plots showing gene expression level distribution of MYC and YBX1 target genes. Control: parental adherent cells.

**Figure 3 cancers-15-02788-f003:**
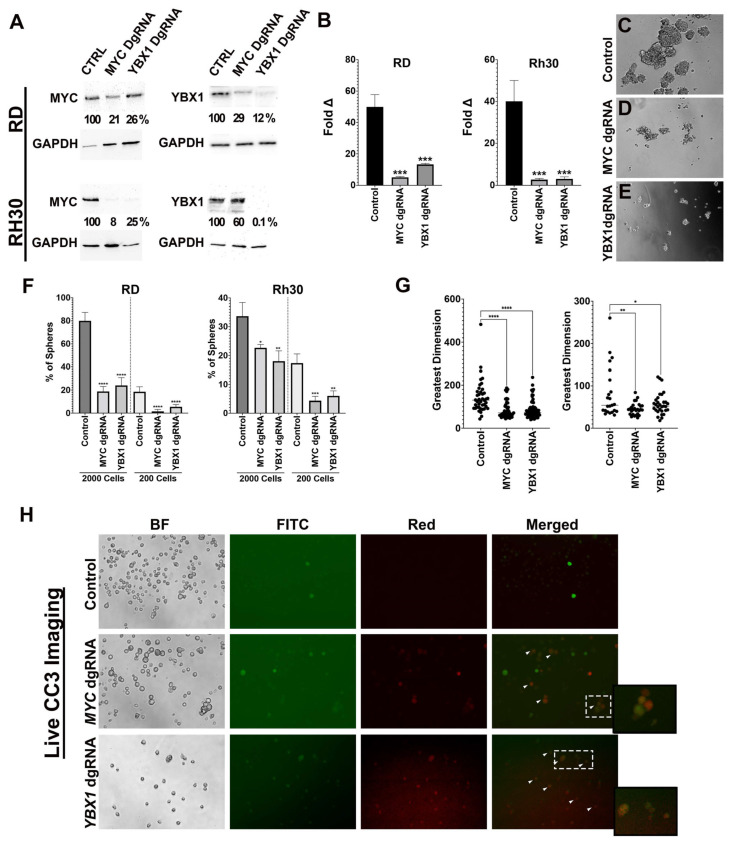
Targeted disruption of *MYC* or *YBX1* reduces RMS tumor cell growth and sphere formation. (**A**) Western blots against MYC and YBX1 in RD and Rh30 cells with CRISPR/Cas9-mediated disruption of *MYC* and *YBX1*. DgRNA = double gRNA. Ratio of band intensity relative to safe-harbor dgRNA control following normalization to GAPDH is shown below each band. (**B**) Cell counts of RD and Rh30 cells with CRISPR/Cas9-mediated disruption of *MYC* and *YBX1*. Stable Cas9-expressing RD and Rh30 cells were transduced with lentiviral CRISPR targeting constructs, and then plated 3 days post-antibiotic selection. Cell counts were performed 6 days later. Cell counts were normalized to day 0. (**C**–**E**) Representative images of RD spheres from CRISPR/Cas9-mediated safe-harbor region targeting (control), *MYC* targeting and *YBX1* targeting. (**F**) Summary of limiting dilution sphere assays for RD and Rh30 cells that were plated at 2000 and 200 cells. (**G**) Summary of sphere size (greatest dimension) quantification as determined by the Image J software, version 64. (**H**) Representative images from live imaging using fluorogeneic NucView^®^ 530 Caspase-3 substrate for detecting apoptotic cells. CD133:GFP SMS-CTR cells were stained with the substrate for 1 h 5 days following lentiviral transduction of the CRISPR dgRNA constructs. BF: bright field. CD133:GFP expressing cells were detected using the FITC channel. The cells labeled with NucView^®^ 530 Caspase-3 substrate were detected using the red channel. Arrow heads in the merged images indicate CD133-positive cells undergoing apoptosis. The inset images show the regions indicated by the dotted lines at higher magnification. For panels (**B**,**F**,**G**), error bar = STD of three replicate wells from one to four independent experiments. ns = not significant, * = *p* < 0.05; ** = *p* < 0.01; *** = *p* < 0.005; **** = *p* < 0.0005 by one-way ANOVA analysis with Dunnett’s multiple comparisons’ test.

**Figure 4 cancers-15-02788-f004:**
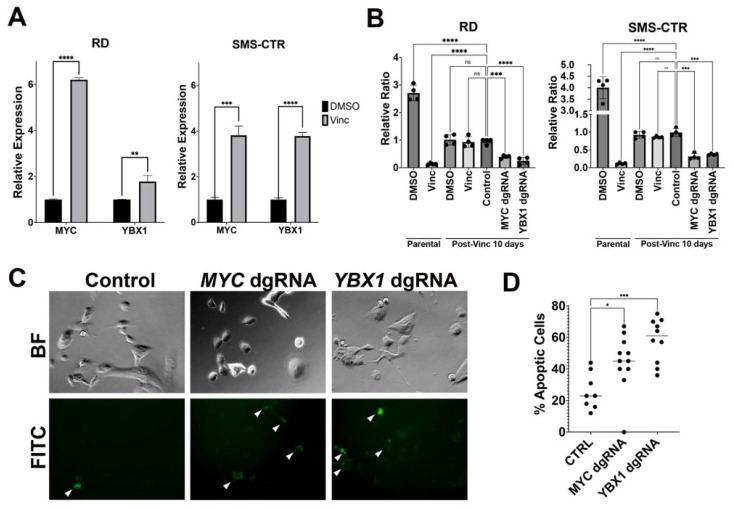
MYC and YBX1 are essential for maintaining viability of vincristine-tolerant RMS cells. (**A**) Summary of quantitative RT-PCR results comparing expression of *MYC* and *YBX1* in RD and SMS-CTR vincristine-resistant cells following 10-day treatment of vincristine at IC 80–90 (1 nM). (**B**) RD and SMS-CTR cells generated from 10-day treatment of vincristine at 1 nM were transduced with lentiviral CRISPR/Cas9 constructs against *MYC* and *YBX1*. Following transduction and selection, cells were exposed again to vincristine continuously for 4 days. Quantitative summary of cell growth analysis from four replicate experiments is shown. Relative ratio in cell counts over 4 days is shown for each condition. Safe-harbor region targeting control was used. (**C**) Representative images from live imaging using the fluorogeneic NucView^®^ 488 Caspase-3 substrate for detecting apoptotic cells in RD cells treated with vincristine for 7 days and transduced with lentiviral constructs for safe-harbor control dgRNA, *MYC* dgRNA and *YBX1* dgRNA. Transduced cells were imaged 8 h after restarting vincristine treatment, and 1 h of exposure to the substrate. White arrows indicate labeled apoptotic cells (**D**) Quantitative summary of the results from the imaging study for apoptosis. For each condition, images were taken from at least 8–10 random fields at 200× magnification across three biological replicates. Each black circle represents the percentage from a field. * = *p* < 0.05; ** = *p* < 0.01; *** = *p* < 0.005; **** = *p* < 0.0005 by unpaired *t*-test in **A**, two-way and one-way ANOVA with Dunnett’s multiple comparisons test in panels **B** and **D**, respectively.

**Figure 5 cancers-15-02788-f005:**
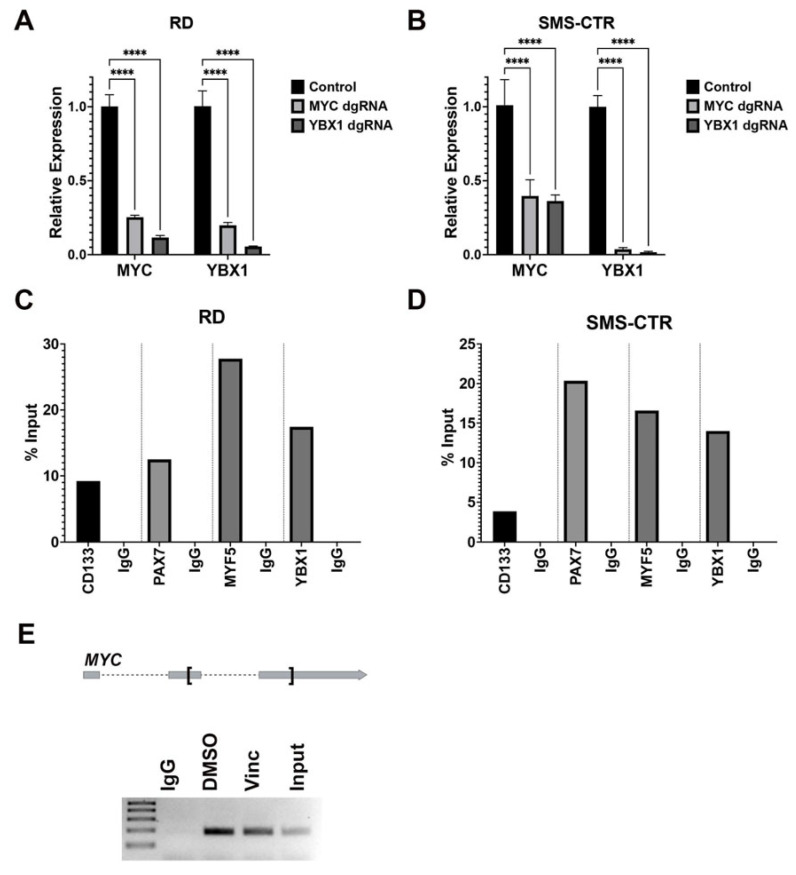
Reciprocal interaction of MYC and YBX1. (**A**,**B**) Summary of quantitative RT-PCR assessing expression of *MYC* and *YBX1* mRNA in RD and SMS-CTR cells with targeted disruption of *MYC* and *YBX1* (8 days post-transduction). **** = *p* < 0.0005 by unpaired *t*-test. (**C**,**D**) Summary of quantitative PCR following CUT&RUN reactions using the MYC antibody against the chromatin extracted from vincristine-treated (**C**) RD and (**D**) SMS-CTR cells. Anti-rabbit IgG was used as the negative control. Shown here is the average of three replicate samples. (**E**) RNA immunoprecipitation assay against YBX1. Shown here is the gel electrophoresis results of quantitative RT-PCR products from RD cells using primers against *MYC* 3′UTR. The regions in *MYC* mRNA amplified by PCR are indicated by the bracket in the schematic at the top of the panel.

**Figure 6 cancers-15-02788-f006:**
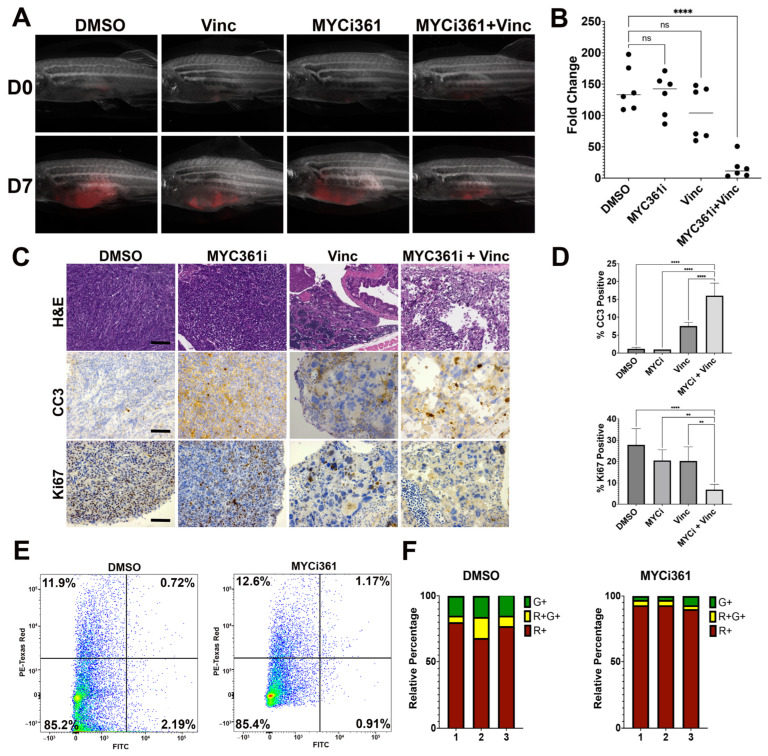
Treatment of zebrafish FN RMS tumors with MYCi361 results in reduced tumor growth and depleted tumor-propagating cell population. (**A**) Representative images of mCherry-labeled KRASG12D-driven FN RMS tumor-bearing CG1-strain zebrafish treated with the vehicle DMSO, MYCi361 (100 mg/kg), vincristine (0.4 mg/kg) and the two-drug combination at day 0 and day 7. (**B**) Summary of tumor volume fold change over the 7-day treatment period. **** = *p* < 0.0001; ns = no significant by one-way ANOVA. (**C**) Representative images of H&E and immunohistochemistry (IHC) for cleaved caspase-3 (CC3) and Ki67 performed on tissue sections from zebrafish ERMS tumors treated with DMSO, MYCi361, vincristine, and the two-drug combination. Scale bar: 500 microns. (**D**) Quantitation of CC3 and Ki67 images. The % of positive cells was quantified over three fields at 400× magnification for each tumor section. The average of six fields total in two tumors for each condition is shown. ** = *p* < 0.01; **** = *p* < 0.0001 by one-way ANOVA with Dunnett’s multiple comparisons test. (**E**) Flow cytometry analysis of fluorescence-labeled cell subpopulations [*myf5*:GFP-positive/*mylz2*:mCherry-negative (G+); *myf5*:GFP-positive/*mylz2*:mCherry-positive (G+/R+); *myf5*:GFP-negative)/*mylz2*:mCherry-positive (R+)] in zebrafish ERMS tumors treated with DMSO and MYCi361 (100 mg/kg). (**F**) Summary of relative ratios of tumor cell subpopulations. Each bar represents one fish.

**Figure 7 cancers-15-02788-f007:**
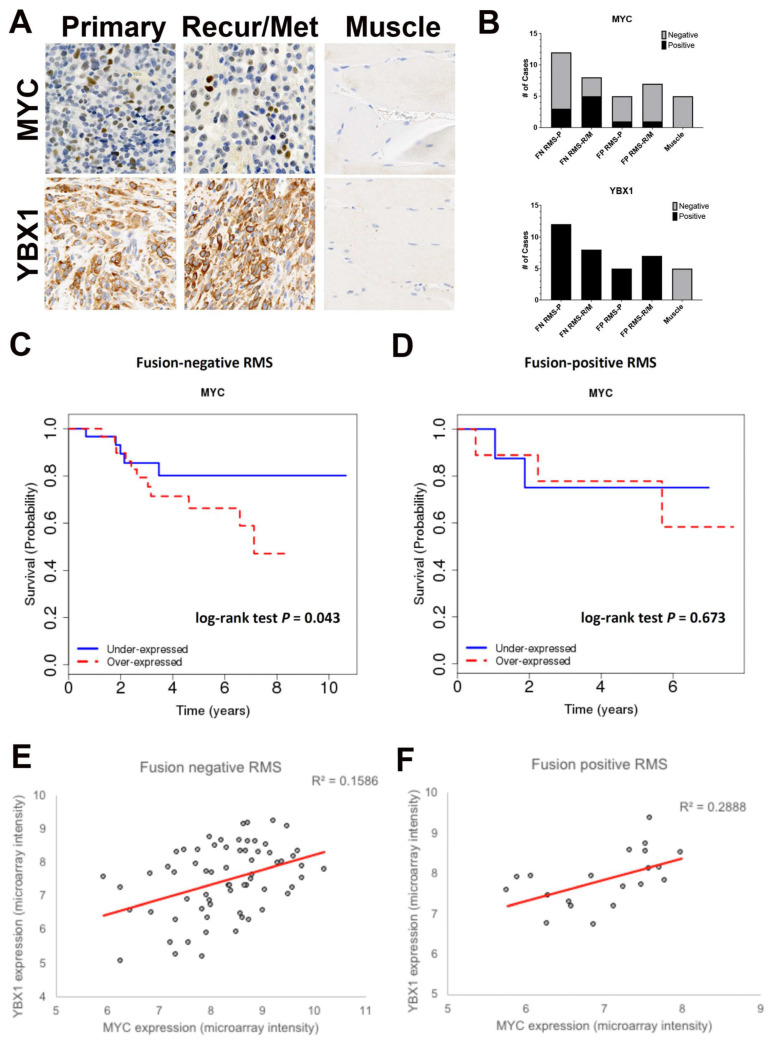
Expression of MYC and YBX1 in RMS with recurrent/metastatic disease and correlation with outcomes. (**A**) Representative images of patient RMS (primary and recurrent or metastatic) and normal muscle tissue sections stained by IHC for MYC and YBX1. (**B**) Summary of IHC results in fusion-positive (FP) and fusion-negative (FN) RMS cases. (**C**,**D**) Pearson correlation analysis on MYC and YBX1 expression levels in FN and FP RMS patients. Correlation coefficient for each comparison is shown. (**E**,**F**) Kaplan–Meier analysis of high and low-expressing MYC FN and FP RMS cases. Each dot represents the value of a single patient.

## Data Availability

The single cell sequencing raw files can be accessed via NCBI GEO under record number GSE232536.

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
