# Peer review of "The MYC-YBX1 Circuit in Maintaining Stem-like Vincristine-Resistant Cells in Rhabdomyosarcoma"

_cancers, 2023, doi:10.3390/cancers15102788_

Round 1
Reviewer 1 Report
In this manuscript was evaluated how MYC and YBX1 affect the viability of chemotherapy-tolerant cells that lead to resistance, which is very important for development of future therapy strategies. I commend the authors for their well-thought-out and comprehensive work
Abstract: The concise results clearly show the most important results of this research
Introduction: The introduction is very well written and covers the most important aspects of RMS strategies in recent years.
Materials and methods: The methodology as well as the extensive experimental work were very well detailed and carried out.
Results: The results were very well described and very well realized and the graphics very well presented.
Discussion: The results were very well discussed and compared with previous results, and I am in full agreement with these conclusions.
I have minor comments:
-The Table S1 is missing
-How was quantification of immunohistochemistry for CD133 on xenograft tumor sections performed? (In legend of Figure S1)
Author Response
We want to thank the reviewer for the positive and constructive feedback. Please see below our responses to the specific comments:
-The Table S1 is missing
RESPONSE: We apologize for the omission. Table S1 is now included in the resubmitted manuscript package.
-How was quantification of immunohistochemistry for CD133 on xenograft tumor sections performed? (In legend of Figure S1)
RESPONSE: We have included the clarification on how the IHC quantification was performed in Figure S1 (please also see below the edits in bold as pasted from the revised supplemental figure legend):
Tissue cross-section images from 3 independent xenograft tumors and spheres were taken at 100x magnification, and positive-staining and total number of cells were quantified using the counter tool in the ImageJ software. The average percentage of CD133-positive cells in 3 representative cross sections from 3 independent xenograft tumors or spheres is shown for each cell line.
Reviewer 2 Report
This article is very well written, and the results and experimental proof is clear. Given that the zebrafish studies are ultimately a useful but non-human model, mouse studies using a human-derived FN RMS cell line or PDX model would be ideal next steps. However, for the current study, the in vivo modeling is appropriate for publication.
No specific edits are suggested, excellent work and well presented.
Author Response
We thank the reviewer for the positive and constructive feedback!
Reviewer 3 Report
This manuscript by Fritzke et al. investigates the cell states and molecular mechanisms in chemotherapy-resistant rhabdomyosarcoma. In particular, they focus on the treatment with Vincristine.
They select for vincristine resistant FN-RMS cells and observe enrichment of stem-like cells supported by upregulation of CD133 and PAX7, OCT4, NANOG and SOX2. The same is observed in FN-RMS spheres. Not surprisingly, CD133 was not enriched in FP-RMS cells selected in the same way, since it is known that FP-RMS possess an intrinsic stem-like phenotype.
They next perform scRNAseq of RMS spheres from two different passages to characterize the subpopulations. 18 sub-populations were identified, but CD133 did not cluster on any particular population. Differential analysis of CD133-high vs. CD133-low RD cells revealed MYC and YBX1 among the top-scored master transcription factors.
The authors very thoroughly validate this finding, demonstrating that both transcription factors are, not surprisingly, involved in RMS cell proliferation and resistance to vincristine, and demonstrate reciprocal regulation.
The most important finding is that treatment of a FN-RMS zebrafish model with the myc inhibitor MYCi361 sensitize the tumors to vincristine.
All in all this is a remarkable work, with very solid experiments and very important finding. The only limitation, which is also addressed by the authors, is how this concept applies to FP-RMS, which follows different rules than FN-RMS. Nevertheless, these finding are very relevant and considerable advance our knowledge on how resistance to chemotherapy arises and how we can counteract.
Minor:
Figure 1L Y-axes legend – Exression -> Expression
Methods : I can’t find the method section for FACS as shown in Fig 1K, antibody used, and methods used to calculate the fold difference.
Line 110 (Fig. S1 E) should be F
Author Response
We thank the reviewer for the positive and constructive feedback. Please see our responses to your specific comments below.
- Figure 1L Y-axes legend – Exression -> Expression
RESPONSE: We apologize for the oversight in the spelling error. The Y-axis label for Figure 1L has been corrected for the resubmission.
- Methods : I can’t find the method section for FACS as shown in Fig 1K, antibody used, and methods used to calculate the fold difference.
RESPONSE: We thank the reviewer for the note. We have included a section on flow cytometry (including sample preparation, gating controls and fold difference calculation) in the Methods section of the revised manuscript.
- Line 110 (Fig. S1 E) should be F
RESPONSE: This has been corrected in the revised manuscript.
Reviewer 4 Report
The authors demonstrated that treatment of RMS cells with vincristine resulted in an increase in CD133-positive stem-like resistant cells. Single-cell RNA sequencing analysis revealed that MYC and YBX1 were among the top-scoring transcription factors expressed in CD133-high cells. Targeting MYC and YBX1 using CRISPR/Cas9 reduced the stem-like characteristics and viability of the vincristine-resistant cells. Additionally, the study found that MYC and YBX1 exhibit mutual regulation, with MYC binding to the YBX1 promoter and YBX1 binding to MYC mRNA.
I have a few comments as shown below:
1. Fig.3A. The GAPDH is not comparable between samples, especially in the upper left panel. It will be difficult to compare the MYC protein levels when the internal control is not comparable in WB, though the authors have the quantitative test.
2. I suggest the authors also provide the sanger sequencing/NGS showing the gene-editing sites are modified in the desired region
3. Though the authors has rescued the phenotype, CRISPR-mediated off-targets and unwanted products should also be involved in the discussion part (PMID: 36639728)
Author Response
We thank the reviewer for the constructive feedback. Please see our responses to the specific comments below:
1. Fig.3A. The GAPDH is not comparable between samples, especially in the upper left panel. It will be difficult to compare the MYC protein levels when the internal control is not comparable in WB, though the authors have the quantitative test.
RESPONSE: We thank the reviewer for the comment. We have attempted the Western blots in 3A multiple times, as both the MYC and the YBX1 antibodies were challenging to optimize on the lysates obtained from the rhabdomyosarcoma cells. The blots shown in 3A are the best ones that we could use for the manuscript. We want to emphasize that the quantitative results of the protein levels on Western blots correlate with our quantitative RT-PCR results on the depletion of the YBX1 and MYC mRNA levels and reciprocal relationship between YBX1 and MYC (Figure 5A). Due to the time constraint on resubmitting the manuscript (5-day turnaround time), we will not be able to attempt additional Western blots to replace the panels in 3A.
2. I suggest the authors also provide the sanger sequencing/NGS showing the gene-editing sites are modified in the desired region
RESPONSE: We thank the reviewer for the insightful comment. We have previously performed PCR using the primers flanking the gRNA sites and demonstrated the DNA deletion events, but the results for which were not included in the initial submission. We previously published the same PCR protocol for checking CRISPR/Cas9-targeted gene deletions (Phelps et al., 2016, PMID: 27956629). We now have included the images of agarose gels showing the deletion bands of expected sizes for MYC and YBX1 in Fig. S4F of the revised manuscript. The purified bands were subsequently sequenced and confirmed to contain the targeted regions. We can attempt to locate the sequencing results if the PCR results as shown in the revised supplemental figure are insufficient for the publication.
3. Though the authors have rescued the phenotype, CRISPR-mediated off-targets and unwanted products should also be involved in the discussion part (PMID: 36639728)
RESPONSE: We agreed with potential background CRISPR-mediated off-targeting events. The discussion on this topic including the citation of the reference as suggested by the reviewer has been included in the paragraph on the limitations of study in the discussion section of the revised manuscript.